# Advances in *Lactobacillus* Restoration for β-Lactam Antibiotic-Induced Dysbiosis: A System Review in Intestinal Microbiota and Immune Homeostasis

**DOI:** 10.3390/microorganisms11010179

**Published:** 2023-01-11

**Authors:** Ying Shi, Jiaqi Luo, Arjan Narbad, Qihe Chen

**Affiliations:** 1Department of Food Science and Nutrition, Zhejiang University, Hangzhou 310058, China; 2Gut Microbiome and Health Programme, Quadram Institute Bioscience, Norwich NR4 7UA, UK; 3Future Food Laboratory, Innovation Center of Yangtze River Delta, Zhejiang University, Jiaxing 314100, China

**Keywords:** antibiotics, *Lactobacillus*, dysbiosis, gut microbiota, metabolites, immunity

## Abstract

A balanced gut microbiota and their metabolites are necessary for the maintenance of the host’s health. The antibiotic-induced dysbiosis can cause the disturbance of the microbial community, influence the immune homeostasis and induce susceptibility to metabolic- or immune-mediated disorders and diseases. The *Lactobacillus* and their metabolites or components affect the function of the host’s immune system and result in microbiota-mediated restoration. Recent data have indicated that, by altering the composition and functions of gut microbiota, antibiotic exposure can also lead to a number of specific pathologies, hence, understanding the potential mechanisms of the interactions between gut microbiota dysbiosis and immunological homeostasis is very important. The *Lactobacillus* strategies for detecting the associations between the restoration of the relatively imbalanced microbiome and gut diseases are provided in this discussion. In this review, we discuss the recently discovered connections between microbial communities and metabolites in the *Lactobacillus* treatment of β-lactam antibiotic-induced dysbiosis, and establish the relationship between commensal bacteria and host immunity under this imbalanced homeostasis of the gut microbiota.

## 1. Introduction

The co-evolution of animals and symbiotic microbes has a history of at least 500 million years, forming a complex but relatively balanced gut microbiota which reach 10^14^ bacteria in a healthy human intestinal tract. Commonly, the gut microbiota of a human consists of 4 dominant phyla, including *Firmicutes*, *Bacteroidetes*, *Protobacteria* and *Actinobacteria*, and more than 500 species [1]. A balanced microbial community possesses relatively stable microbial metabolites and surface antigens, which are crucial for immune system maturation and responses. Gut microbiota perturbation is generally observed in diseases relating to metabolic or immune disorders [2,3]. Therefore, it is important to clarify and examine the interactions between microbiota and host diseases. Through the characterization of microbial species and the whole microbial communities, the effects of microbial metabolism on the host’s immune system could be further revealed.

Distinct antibiotics exhibited diverse inhibition spectra, and the inhibitor mechanism of the β-lactam antibiotic was more mild than the killed ones. Therefore, antibiotic treatment results in the enhanced susceptibility of the host to pathogen infection and the development of aberrant immunity, as the presence of a healthy microbiota is crucial for the prevention of the colonization of enteric pathogens [4,5]. The dysbiosis state may cause further host diseases, hence, it is important to understand how the alteration and potential defects of the microbial community, metabolites and intestinal mucosa immunity are linked.

*Lactobacillus* strains are resident in the gastrointestinal tract and essential for the fermentation of cheese, wine, yoghurt and other brewing foods. The beneficial effects of *Lactobacillus* in the restoration of antibiotic-induced dysbiosis have been reported recently. A series of large-scale, double-blind, well-designed trials were conducted in validation of *Lactobacillus’s* effect on antibiotic usage. It is difficult to confirm the efficiency of *Lactobacillus* strains due to the diversity of strain-dependence, dosage of use and appropriate style. Therefore, we first discuss the effects of antibiotics on the host’s gut microbiota, homeostasis and the causes of dysbiosis. Second, we review studies elucidating the role of *Lactobacillus* under β-lactam antibiotics in gut dysbiosis and the related metabolic disease. Finally, we examine how *Lactobacillus* acts in dysbiotic microbiota and a disturbed immune system, as well as the interaction between metabolites and altered immunological pathways under antibiotic exposure (Figure 1).

## 2. The Pathogenesis of Antibiotic-Induced Dysbiosis and Gut Microbiota Dysfunction

### 2.1. The Antibiotics Cause Diverse Gut Bacteria Community and Destroy Homeostasis

Distinct inhibition of antibiotic spectra varied from β-lactams spectra. Macrolides and tetracyclines are both prototypic bacteriostatic protein synthesis inhibitors, and they inhibited or killed nearly all the tested commensal species. This species-specific killing activity challenges the long-standing distinction between bactericidal and bacteriostatic antibiotic classes and provides a possible explanation for the strong effect of macrolides on animal and human gut microbiomes.

The human gut microbiome comprises trillions of microorganisms, which are mostly bacteria and mainly considered as non-pathogenic. As a “digestive organ,” the gut microbiome, together with the intestinal tract, has intimate interactions and co-evolved mutualistic relationships [6]. The intestinal tract offers a rich environment for the growth of a microbial community, and the microbiome performs a crucial role in the metabolic and immune systems to protect against resident opportunists and pathogen invasion [7]. Indeed, the microbiota also play an important role by providing essential vitamins, amino acids and short-chain fatty acids (SCFA) [8]. The cause of disease is generally related to the dynamic between the host immunity and the pathogens. Recently, the alteration of resident microbiota in patients with a microbial infection has revealed new aspects of pathogen biology. However, further verification is needed to accurately identify the consequences of the gut microbiome on human health and disease.

Recently, the incidence of numerous human gut, metabolic or immune diseases, including infection, inflammatory bowel disease (IBD), colorectal cancer, allergy, diabetes, obesity, autism and asthma, has extensively increased, and these diseases have been connected with the change in microbiome structure termed dysbiosis [9,10,11,12,13,14]. Due to the diverse functions of the gut microbiome, the maintenance of a balanced gut is also crucial for intestinal homeostasis and health. It is essential to understand the mechanisms that lead to dysbiosis and whether the modulation of gut microbiota can provide positive clinical outcomes. Immune homeostasis, which is intimately related to microbiota composition, is maintained and achieved because of the extensive interaction between the microbial community and mucosal immune system. The human immune system needs to establish a suitable balance between the vigilance to guard against infectious pathogens and tolerance of the commensal microbiota. The interplay between the host and microbiota is extensive, thus gut homeostasis is maintained as an inflammatory tone, which allows for an appropriate response to infectious agents or stress.

### 2.2. Antibiotic Treatment Induces Alterations in the Microbial Community and Metabolic System

Dysbiosis is a distinct microbial ecological state which is causally linked to the usage of exogenous materials or the treatment of disease [15]. Although each individual host harbors a relatively constant gut microbiota over time, which represents normal intestinal microecology, such a particular status can be modified by several factors, such as infection, the usage of antibiotics, dietary alterations or inflammation [16]. These shifts in the relative abundance of microbial species are referred to as gut dysbiosis, which is not only related to the altered composition of gut microbes, but also associated with functional changes in the microbial metabolome, transcriptome or proteome.

#### 2.2.1. Loss of Commensal Bacteria

The reduction or complete loss of a commonly resident bacterial community caused by microbial diminishing or decreased bacterial proliferation could induce gut dysbiosis [17]. Such a loss of commensal bacteria may be important to intestinal homeostasis and functions, in which case, the supplementation of the lost bacteria or related metabolites could be utilized as a strategy to restore dysbiotic-associated phenotypes.

#### 2.2.2. Alteration in Abundance of Specific Taxa

The altered abundance of some potentially beneficial microbes or opportunistic bacteria could affect the overall microbial community and lead to changes in particular microbiota-derived metabolites [18]. This has been demonstrated recently, for example, by the relative abundance of *Enterococcus* blooms in an antibiotic-induced dysbiosis model, which may drive the pathogenesis of experimental gut inflammation and human IBD [19]. In this dysbiosis model, the decrease in *Akkermansia* was also proven to be a biomarker for an unhealthy gut [20] and was correlated with the incidence of ulcerative colitis. In addition, the emerge of *Escherichia coli* in antibiotic-treated mice induced bacteremia and mortality was demonstrated [21].

#### 2.2.3. Reduction of Diversity

The reduction of α-diversity, which represents the species richness, is commonly associated with dysbiosis. Generally, the richness of gut microbiota increases during the first year for newborns [22], and is influenced by diet, birth mode, antibiotic use and other factors [23]. It has been reported that antibiotic-treated children had relatively less stable microbial communities and less diverse gut microbiota [23]. In addition, the lower diversity of microbiota which is related to dysbiosis was induced by an abnormal diet, type 1 diabetes or an immunodeficiency syndrome [18,24,25].

#### 2.2.4. Induction of the Intestinal Microbial Dysbiosis

Antibiotics are conventionally bactericidal or bacteriostatic, and indiscriminately restrict or kill both pathogenic and beneficial bacteria. It has been documented that antibiotics affect the genomic, taxonomic and functional characteristic of the microbiome, and they could induce either short-term or everlasting consequences [26]. The altered diversity and abundance of the microbiota after the usage of antibiotics can lead to an increased risk of colonization by pathogenic bacteria, which occasionally leads to the expansion of *Clostridium difficile*, particularly in the elderly population, and further causes diarrhea or fatal colitis [27].

Antibiotics are classified into several types, including β-actin, fluoro-quinolone, glycylcycline, lincosamide, nitro-imidazole, a combination (β-lactam/amino-glycoside) and others. The β-actin antibiotic-induced dysbiosis of the gut microbiota has been listed in Table 1. The co-evolved microbiota, which developed together with the host, is altered by antibiotic treatment, with significant consequences for the host’s health. Antibiotic-induced consequences on the microbial regulation of the host mainly include the dysbiosis of microbial communities, the loss of bacterial ligands, changes in metabolites and alterations of bacteria-directed immune signals.

#### 2.2.5. Alterations in Microbial Metabolites

After antibiotic treatment, one of the common features of metabolic profiles in mice and humans is the decreased level of SCFA. SCFA, including acetate, propionate and butyrate, are normally produced by bacteria through undigested carbohydrates, then absorbed by host cells in the colon. After their consumption by colonocytes, the remaining SCFA are transported to the liver through the blood stream. They also play a role as signaling molecules in the regulation of glucose and lipid metabolism [35]. According to the previous studies, the two G-protein-coupled receptors, including GPR41 and GPR43, which are renamed as FFAR3 and FFAR2, respectively, are the main intermediaries involving the metabolic interactions, and these two receptors are broadly distributed in the organs and tissues of humans, especially in the small intestine and the colon [36,37].

Two independent studies with cefixime- and cefoperazone-treated mice showed a reduced concentration of SCFA or free fatty acids, secondary bile acids and glucose, respectively, indicating that the microbe-related metabolic function was disrupted [32,34]. In accordance with these observations, some taxa of microbiota were not restored after treatment with antibiotics, and several metabolites had not recovered even a few weeks after the cessation of antibiotic treatment [34]. The decreased level of SCFA can be crucial to gut health and immunity, some of which also influence the differentiation and apoptosis of cells and provide energy sources for the gut epithelium.

### 2.3. Antibiotic-Induced Ecological Dysfunction and Related Gut Disease

Recent studies have documented that antibiotics have short- or long-lasting alterations in the gut microbiota and related diseases. Antibiotic use for 1 week or less in healthy humans was reported to affect the diversity of bacterial taxa, deplete the specific families or genera and up-regulate the antibiotic resistance genes, and these effects persisted for 6 months to 2 years after the cessation of antibiotic intake. Furthermore, the antibiotic-induced microbiota perturbation may lead to disease in both infants and adults, especially those correlated to the allergic or metabolic syndromes. One study revealed a relationship between the usage of antibiotics during early life and the incidence of asthma by the seventh year based on data from thousands of children. As with the human study, mice studies also demonstrated that neonatal exposure to antibiotics induced more severe symptoms of asthma. Accordingly, penicillin administration during the gestation period resulted in more enhanced physiological changes than during the weaning period, indicating the important role of microbiota inmaintaining health during early life [28]. Therefore, short- or long-term exposure to antibiotics during infancy and at an adult age could significantly influence the composition and abundance of the microbiome, which could further result in metabolic- or immunity-related diseases in the host.

#### 2.3.1. Dysbiosis and Inflammatory Bowel Disease (IBD)

As the most frequent forms of IBD, the mechanism of ulcerative colitis (UC) and Crohn’s disease (CD) remains unclear, however, extensive evidence shows that intestinal microbial dysbiosis acted as a major inducer of IBD [38,39,40,41]. The observed microbiota dysbiosis in IBD patients could potentially be linked to the occurrence and severity of this disease [40]. The dysbiotic microbiota has proven to be related to the alteration of *Dialister invisus*, *Bifidobacterium adolescentis*, *Clostridium cluster* XIVa, *Faecalibacterium prausnitzii* and *Ruminococcus gnavus* in CD [42]. *Faecalibacterium prausnitzii* is especially connected to the anti-inflammatory effects of the disease [43]. However, whether gut dysbiosis is the main cause of the inflammatory reactions in IBD remains obscure.

#### 2.3.2. Dysbiosis and Other Gut Disorders

Apart from IBD, the dysbiotic state of the gut was related to other gut diseases such as coeliac disease, irritable bowel syndrome (IBS) and colorectal cancer. Earlier studies indicated significant differences in the diversity of the microbiotas between IBS and healthy individuals. Other studies demonstrated that the microbial communities of *Proteobacteria* and *Firmicutes* were increased and the abundance of *Lachnospiraceae* was enriched in the diarrhea-predominant IBS (D-IBS) patients, indicating that significant differences exist in the specific bacterial taxa between D-IBS patients and healthy human [44]. The dysbiosis microbiota was detected in coeliac disease, including the reduced abundance of *Bifidobacterium* and *Clostridium histolyticum;* in addition, the level of IgA-coated bacteria was also linked to such gut dysbiosis [45]. The characterization of mucosal bacteria in colorectal cancer patients revealed that the alterations in microbial communities related to adenomas may exacerbate the colorectal cancer [46]. The relationship between microbiota composition and host genetics was correlated with coeliac disease development, in which an altered microbiota composition was observed together with the increased expression of leukocyte antigen DQ2 in coeliac disease children [47]. However, the consistent pattern or feature of bacteria alterations in these diseases has not been concluded.

#### 2.3.3. Dysbiosis and Metabolic Disorders

Recently, the gut microbiota dysbiosis has also been associated with metabolic diseases such as type 2 diabetes and obesity. The microbiota was changed in the obese mice, however, the incidence of metabolic endotoxemia and obesity-related parameters was reversed after antibiotic-conferred changes in the microbiota. Early-life antibiotic exposure can induce dysbiosis of the gut microbiome in fat-1 mice and lead to later-life obesity [48]. Indeed, the levels of SCFA related to the gut microbiota, as well as the alterations in microbial communities, were crucial in the development of obesity. It has also been demonstrated that the supplementation of a high-fat diet with SCFA protected diet-induced obesity via the peroxisome-proliferator-activated receptor (PPAR)-dependent switch [49]. In addition, the obesity was proven to be linked with intestinal permeability, which is also associated with altered gut microbiota [50]. A moderate shift in the microbial community was observed in the type 2 diabetes, and the decrease of some butyrate-producing bacteria was also detected [51]. Although the intestinal microbial markers were characterized to demonstrate the classification of type 2 diabetes, whether the dysbiosis of gut microbiota is the cause of metabolic disorder diseases still needs to be proven. The related studies showed debatable results for this question, that the composition of the gut microbiota was adapted to the diet changes in obese individuals [52] while the transferred microbiota from lean donors to the metabolic syndrome individuals could attenuate the metabolic symptoms [53].

### 2.4. Antibiotic-Induced Aberrant Immune Response and Inflammation

The activation of microorganism-associated molecular pattern (MAMP) recognition receptors is the crucial mechanism by which microbiota can affect host immunity, and these include Toll-like receptors (TLRs) and NOD-like receptors (NLRs). Some intestinal ligands modulated by microorganisms including LPS, peptidoglycan and lipoteichoic acid can activate pattern recognition receptors (PRRs). The activation of these ligand-engaged receptors on the epithelium can also result in proinflammatory responses [54]. The microbial ligands stimulate the nuclear factor kappa B (NF-κB) and proinflammatory cytokines TNF-α. The depletion of the gut microbiome by antibiotic exposure results in reduced numbers of MAMPs in the epithelium, decreased TLR signalling and down-regulated innate defenses [55]. The microbiota plays an important role in assisting the mucosal immune system in its defense against infection. The use of germ-free animal models has revealed the crucial consequences for the microbiome and immune system, including lymphoid tissue development, T cell, antimicrobial peptides and secretory immunoglobulin A (sIgA) [56]. Enteric infection-induced dysbiosis was initially observed in the infection with *Citrobacter rodentium,* where the microbiota lacked its ability to resist invasion by pathogenic bacteria (Figure 2).

In a normal microbiota environment, the homeostasis of the microbial community and host are sustained in relatively stability. The recognition of the microbiota signals through the NF-κB pathway and induces the production of AMPs. The expression of tight junction proteins and the levels of metabolite are relatively constant.In antibiotic-induced dysbiosis, the numbers of MAMPs are decreased as a result of the loss of microbiota, and the thinned mucus layer and damaged tight junction proteins lead to the infection of opportunistic pathogens. The levels of SCFA and related immune functions are altered after antibiotic exposure.

Furthermore, the studies of antibiotic use have proven that the gut microbiota also contributes to the maintenance of the immune system. Antibiotic treatment aggravated the DSS-induced colitis of mice by eliminating the microbial ligands which sustain the gut homeostasis through signaling TLRs. Amoxicillin administration resulted in the appearance of antigens and the decreasing secretion of antimicrobial defense molecules, such as phospholipase A2 and α-defensins. The loss of specific taxa of the microbial community with antibiotics could identify which microbial taxa may be responsible for the regulation of T cell activation. For example, the ampicillin treatment targeting the Gram-positive bacteria induced the depletion of the T_H_17 cell population [29].

The related immune impairment occurred after antibiotic-induced dysbiosis. After a low dose penicillin treatment, the expression of immune genes for Reg3γ, IL-17 and β-defensins was decreased in dysbiotic mice [28]. Reg3γ is a bactericidal C-type lectin whose basal level is sustained through bacteria-derived LPS that presents in Paneth cells and intestinal epithelial cells (IECs), hence Reg3γ could not be detected in germ-free mice. The antibiotic treatment was reported to influence the expression of Reg3γ, and the oral supplementation of LPS can reverse this deficiency [57]. Moreover, the neutrophil numbers and production of IL-17 was reduced after antibiotic exposure, which is also associated with an increased abundance of *Escherichia coli* and *Klebsiella pneumoniae* sepsis [58]. The antibiotic-treated disturbance induced the basophil development through the Th2-IL-4-IgE pathway and resulted in an increase in inflammation and allergic diseases in mice [59]. Similarly, after the antibiotic treatment, the levels of T cells, immunoglobulin, interferon γ (IFNγ), IL-18 and IL-1β were increased and dendritic cell (DC) migration was activated [19,60], indicating that microbial-modulated signals contribute to the systemic immunity.

In addition, antibiotic usage was shown to induce the reduction of CD8^+^ T cell expansion, with levels of IgG, IFN-γ and TNF-α resulting in immunity injury in mice [61]. The exposure of the mice to amoxicillin and clavulanic acid also induced the decrease of IgG in serum. The interaction between the microbiome and immune system is influenced by antibiotic treatment through the regulation of T lymphocyte activity. For instance, antibiotic treatment which targets Gram-positive bacteria could affect the development of Th17, resulting in a SFB-lacking microbial community and the susceptibility to infection by *Clostridium rodentium* [29]. A 10-day antibiotic treatment led to the reduction of CD4^+^ T cells, IFNγ and IL-17, indicating that microbial signaling is essential in the maintenance of T cell effectors [62].

## 3. Modulation of Antibiotic-Induced Microbial Dysbiosis by *Lactobacillus*

### 3.1. Restoration Effects of Lactobacillus after Antibiotic-Induced Dysbiosis in Animals

In recent decades, the intake of *Lactobacillus* strains and other probiotics has applied in the therapy or intervention of antibiotic-induced dysbiosis [63], and these studies using *Lactobacillus* have investigated the mechanisms leading to the amelioration of antibiotic-induced dysbiosis in humans and animals (Table 2). The dose of oral *Lactobacillus* strains was varied from 10^8^ to 10^11^ CFU, and 10^10^ CFU were applied for administration in most studies.

In a murine model, the introduction of *L. paracasei* recovered abundance of phylum Bacteroidetes in vancomycin-resistant enterococci persistence [66], which could have effect on intestinal host response through potentially creating a more favorable niche. Another experiment using cocktails of *L. casei*, *L. plantarum*, *L. rhamnosus* and *L. helveticus* demonstrated that *Lactobacillus* treatment enriched the abundances of *Akkermansia* and *Porphyromonadaceae* while reducing the populations of *Sporobacter*, *Robinsoniella*, *Oscillibacter*, *Ruminococcus*, *Clostridia* and *Helicobacter* in antibiotic dysbiotic state [19,79], which are partly responsible for maintaining immune homeostasis. Similarly, ampicillin-exposed mice suggested that *Lactobacillus* administration contributed to the promotion of a more stable gut microbial community by reduction of *Klebsiella* and *Enterococcus* and the enhancement of *Coprobacillus*, Bacteroidales and *Eubacterium* in all three *Lactobacillus* treatment groups [65].

Importantly, the strain-specific effects on the antibiotic-induced dysbiosis were also evaluated amongst a number of studies. Recently, *L. casei* CGMCC 12435 (LacC) strain was proven to promote specific bacterial taxa including *Citrobacter*, *Bifidobacterium* and levels of SCFAs to attenuate ampicillin-induced dysbiosis while other two tested strains (LacP and LacG) had no such effect, suggesting strain-specific restoration of microbial community. Another study reported that *L. reuteri*, but not *L. rhamnosus* GG or nonpathogenic *Escherichia coli*, decreased the antibiotic-induced elevation of the Firmicutes: Bacteroidetes ratio. These differences and mechanism of *Lactobacillus* strains in enhancement of gut microbiota disorders and regulation of intestinal permeability could be further investigated.

Various studies have indicated that decreased diversity of gut microbial community could lead to the development of gut dysbiosis or other diseases [80,81,82,83]. Single or mixture *Lactobacillus* strains are presumed to benefit intestinal health by their relatively direct actions on the composition and function of gut microbiota. Previous studies conducted effects of single strain either *L. reuteri* 6475 or *L. rhamnosus* GG, and *L. reuteri* but not *L. rhamnosus* reduced the ratio of Firmicutes: Bacteroidetes elevated after antibiotics use. Also, a modulated intestinal permeability and reduced femoral trabecular bone volume were observed [67]. Subsequently in an Apis mellifera model, mixture of three immunostimulatory *Lactobacillus* strains mitigates both head and gut antibiotic-associated microbiota dysbiosis in adult bees by suppressing larval pathogen loads to near-undetectable levels [68]. Despite multiple reports noting that cocktail or single *Lactobacillus* consumption improved and recovered the gut microbiota composition toward pre-treatment state while some studied had no obvious results in alteration of the diversity of gut microbial community [84,85], the potential mechanism of gut microbiota restoration by *Lactobacillus* in microbial niches remains in need of discussion and exploration.

### 3.2. Roles of Lactobacillus Strains in Clinical Antibiotic Treatments

The antibiotic-induced dysbiosis in human body is frequently accompanied with antibiotic-associated diarrhea (AAD). Both clinical and routine interventions were investigated to analyze the effects of *Lactobacillus* during or after antibiotic administration. *L. reuteri* had been performed for 28 days in 31 patients receiving antibiotics, and decreased the incidence of AAD among hospitalized adults [69]. Consumption of *L. plantarum* has been proven to minimize the risk of developing loose or watery stools and a similar effect was observed in application of *L. acidophilus* NCFM [71,73].

However, it should be emphasized that *Lactobacillus* strains exhibited obvious alleviation in antibiotic-induced moderate or mild gut dysbiosis while these effects were weakened or disappeared in the severe cases. For example, when *L. rhamnosus* GG or *L. casei* Shirota was administered respectively in hospitalized severe patients receiving antibiotics, no restored improvement was observed in gut dysbiotic symptoms [75]. On the contrary, a quite different outcome of *L. rhamnosus* GG or *L. paracasei* Lpc-37 were demonstrated that patients with an initial mild to moderate antibiotic administration could be alleviated in the gut dysbiosis [73]

Both single and cocktail *Lactobacillus* strains were broadly applied in the antibiotic-induced dysbiosis. Cocktail strains including *Lactobacillus*, *Bifidobacterium*, *Lactococcus* has been demonstrated to improve microbiological, clinical and immunomodulatory efficacy in case of post-antibiotic reconstitution of the gut mucosal [72]. Also, mixture of *L. acidophilus* and *L. casei* has been shown to be effective against antibiotic-associated diarrhea caused by *Clostridium* infections [74]. Meanwhile, a combination of *L. helveticus* and *L. rhamnosus* supplementation has also supported its efficacy in reducing the duration of diarrhea symptoms [86]. Another mixture of two *L. reuteri* strains could reduce the antibiotic-induced side effects, including dysbiosis and stomach problems [76]. On the other aspect, single strain administration of *Lactobacillus* was also broadly studied. Several reports including *L. reuteri* or *L. plantarum* 299v have been proven to alleviate antibiotic-induced dysbiosis, while a few studies revealed that *L. casei* Shirota or *L. rhamnosus* GG had no specific improvement on the antibiotic-induced side effects [71,75].

Based on this research, the restoration of *Lactobacillus* cocktails in antibiotic-induced dysbiosis is more obvious than the effects of a single strain, however, the mechanism of these differences between single and mixture strains has not been investigated and remains unclear.

## 4. The Modulation Mechanism of *Lactobacillus* in Gut Microbiota and Immune Systems under Antibiotic-Induced Dysbiosis

### 4.1. Restoration of Microbial Community Composition by Lactobacillus Strains

The alterations of gut microbiota are relatively direct impacts caused by antibiotic administration. Therefore, the improved effect of *Lactobacillus* strains on gut microbiota rebuilding is one of the essential aspects of modulation evaluation. Most studies focus on two points, which are the promotion of comparatively beneficial microbes and the suppression of unexpected bacteria or pathogens. For instance, the treatment of *Streptococcus boulardii* CNCM with a seven-day administration of amoxicillin-clavulanate reduced the levels of *Escherichia coli* overgrowth [87]. Moreover, a mixture of *L. acidophilus, Bacillus fragilis* and *Bifidobacterium longum* treatment enhanced the relative abundance of *Akkermansia* [88]. Especially, some *Lactobacillus* could selectively protect against *Bacteroides* species but not against related pathogens in antibiotic-induced dysbiosis [89]. These findings suggest *Lactobacillus* strategies for restoring the gut microbiota specifically and illuminating the activity spectra of β-lactam antibiotic in resident bacteria.

Importantly, although an antibiotic-induced dysbiotic gut microbiome can be partly or totally reversed by *Lactobacillus* strains, the residual impact on microbial metabolism and the immune system may persist for a long time, and the negative effect of *Lactobacillus* strains could also exist in some severe antibiotic-induced cases [90]. In a short course of clindamycin administration, the profound loss of microbiota by one-third elimination was relatively hard to restore using *Lactobacillus* treatment [91]. Especially, it was also reported that the disturbance of antibiotic-induced gut dysbiosis can extend all the way down to the genetic level and re-shape the microbiota composition, although the ecological resilience of the microbiota could be mildly restored to the baseline levels [92,93]. The varying impact of antibiotics on subjects is associated with differences in doses, lifestyles, demographics and analysis methods [94]. Therefore, the administration of antibiotics potentially affects the evolution and ecology of the human gut microbiome, and further generates wide-ranging implications for the host’s health (Figure 3).

### 4.2. Modulation of Gut Barrier Function and Intestinal Immune Response by Lactobacillus Strains

Antibiotic exposure is one of the key factors that induces gut barrier disruption and immune dysfunction, and their overuse can affect gut barrier integrity and aggravate various health issues [95]. The maintenance of intestinal barrier integrity has been proven to be crucial in the homeostasis of gut microbiota and the prevention of antibiotic-induced diseases [96]. As previous reported, antibiotic administration can cause damage to the gut barrier, which is mentioned as leaky gut, and leads to a series of immune responses and metabolic disorders [97]. Antibiotic-induced disruption to the barrier function of intestinal epithelium increases the permeability of the mucosal lining, exposing it to allergens, pathogens and toxins. In recent studies, *Lactobacillus* strains could colonize or promote the growth of beneficial microbes in the anaerobic intestinal tract via the formation of biofilms and the enhancement of gut mucosa. In addition, *L. plantarum* has been reported to secrete plantaricin EF and prevent damage to the gut barrier integrity through LPS-repairing [98]. Furthermore, gut barrier dysfunction increases gastrointestinal infections and may result in bacterial translocation to intestinal mucosa; accordingly, *Lactobacillus* strains can restore antibiotic-induced gut barrier disruption through the inducible expression of tight junction proteins. For instance, previous research announced that *Lactobacillus* cocktails may enhance the levels of tight junction proteins, including ZO-1 and occludin, and such a rearrangement leads to an immune response [33].

In recent years, several therapy strategies and alternative applications have been suggested to alleviate antibiotic-induced dysbiosis and mitigate the side effects of antibiotics. Several new therapies focus on the intervention of probiotics, including *Lactobacillus* or prebiotics, and their beneficial effects on intestinal microbiota and immunity. Summarily, the majority of research on this subject has proven that *Lactobacillus* administration can be impactful in antibiotic-induced dysbiosis. The administration of *Lactobacillus* can modulate and produce a positive impact on the microbial community through the regulation of the immune response, including the production of anti-pathogen inhibitory substances and the blockage of adhesion sites [99,100]. It was reported that the administration of *Lactobacillus brevis* could generate an effective immune response in breast cancer mice. Importantly, in *L. casei*, Zhang decreased the serum IL-1α level, indicating inflammation reduction after antibiotic treatment [101], however, increased systematic inflammatory responses and gut antibiotic resistance genes are relatively hard to recover with *Lactobacillus* treatment.

Moreover, the regulation of Treg expression induction through CD4^+^Foxp3^+^ T cell differentiation has been revealed by *Lactobacillus* strains, which suggests the importance of Treg cells in maintaining gut homeostasis. Relevantly, antibiotic-induced gut barrier disruption accelerated the translocation of pathogen evasion, and therefore increased pathoadaptive mutations of commensal intestinal microbiome genes and resulted in infective immune responses or bacterial dissemination [102]. *Lactobacillus* species targeted regulated immunity cytokines and suppressed the potential invasion of pathogenic bacteria, which created a new strategy for restoring the pathways of beneficial species.

### 4.3. The Metabolite (Including SCFAs)-Sensing Related Immune Response after Antibiotic Treatment

*Lactobacillus* strains release metabolic molecules such as short-chain fatty acids (SCFAs) that modulate immune responses and up-regulate the expressions of tight junction proteins. The SCFA have been reported to be involved in the development of colonic regulatory T (Treg cells) and engage the G protein coupled receptor GPR43 on neutrophils to decrease their infiltration into tissues and alleviate inflammation, thus SCFA could also enhance the expression of FoxP3 and drive the naïve T cell to Tregs [103,104]. It has been proven that antibiotic exposure significantly decreased the concentrations of SCFA in the faeces of mice [32], and the administration of SCFA promoted the T cell-induced colitis of mice through enhancing the expression of Tregs [103]. Indeed, SCFA were proven to induce the production of cathelicidin-related antimicrobial peptides and hence promote the inducement of Tregs [105]. The antibiotics targeting Gram-positive bacteria affect signaling through NOD2 and TLR-2, while those targeting the Gram-negative bacteria sense signals through the NOD1 and TLR-4 pathways [55].

In addition to their activity as the ligands for G protein-coupled receptors (GPCRs), SCFA also act as the inhibitors of histone deacetylases (HDACs). HDAC inhibition produces anti-inflammatory response via the interaction of SCFA with DC and macrophages [106,107]. Hence, SCFA-induced HDAC inhibition is important in the regulation of innate immune responses and NF-κB activity. SCFA regulate signaling through GPCRs, including GRP43, GPR41 and GPR109A, which are expressed by IECs and immune cells. The GPR109A is a receptor that responds to butyric acid, which prevented colitis by increasing the expression of anti-inflammatory cytokines and inducing differentiation of T_reg_ cells [108]. Collectively, microbial-derived SCFA are involved in regulating inflammation to affect immune responses and maintain mucosal homeostasis. SCFA also contribute to the tight junction barrier of IECs. The enhanced level of acetate was shown to decrease the risk of *Escherichia coli* O157:H7 infection, promote the IEC integrity and further inhibit the translocation of Shiga toxins [109].

As reported, more than 70 gut bacteria have been investigated to produce SCFAs, most of which belong to the well-studied probiotic genera *Lactobacillus*, *Bifidobacterium* and *Clostridium*. Most of these genera are safe and retain high adaptability to the gut ecosystem. In summary, the antibiotic-induced gut microbiota dysbiosis causes the decrease of SCFAs, which further results in the alteration of the regulation of the immune system. For instance, acetate acts as a substrate for fatty acid or cholesterol synthesis, which increases the rate of oxygen uptake and the movement of ileum by affecting its contraction [110,111]. The regulative effects of *Lactobacillus* by metabolites targeted two main aspects; one is the production or synthesis of functional SCFAs and other metabolites by specific *Lactobacillus*, and the other is the affection of relevant microbial species by *Lactobacillus* that enhance levels or ratios of beneficial metabolites.

## 5. Conclusions and Future Perspectives

Here, we have highlighted the effect of antibiotic exposure in the disruption of commensal communities, and the resulting decreased interactions between gut microbiota and the mucosa. Based on a number of studies, we summarized the alteration of the appropriate metabolites and functional immune system in a host with antibiotic-induced dysbiosis. The antibiotic treatment has long- or short-lasting perturbations in microbiota that result in the potential risk of diseases including gut disorders, metabolic syndromes and immune-related defects. Therefore, treatments of *Lactobacillus* aimed at restoring the microbial community and gut barrier integrity are likely to directly reshape the gut ecosystem. Instead, we believe that intestinal metabolic alterations involving the consideration of SCFAs, immune regulatory factors and signaling receptors are more likely to show positive results in *Lactobacillus*-involved antibiotic dysbiosis regulation. However, we urgently need alternative strategies that substitute or supplement the currently widespread use of antibiotics.

Recent advances should offer further evidence related to the unascertained association between animal experiments and the clinical efficacy of *Lactobacillus* in antibiotic-induced dysbiosis. Furthermore, the effectiveness of *Lactobacillus* depends on the severity of the antibiotic-induced disease and specific function on the strain level. Indeed, the treatment between a single strain and mixture of different *Lactobacillus* genera or species should be urgently investigated in future studies.

## Figures and Tables

**Figure 1 microorganisms-11-00179-f001:**
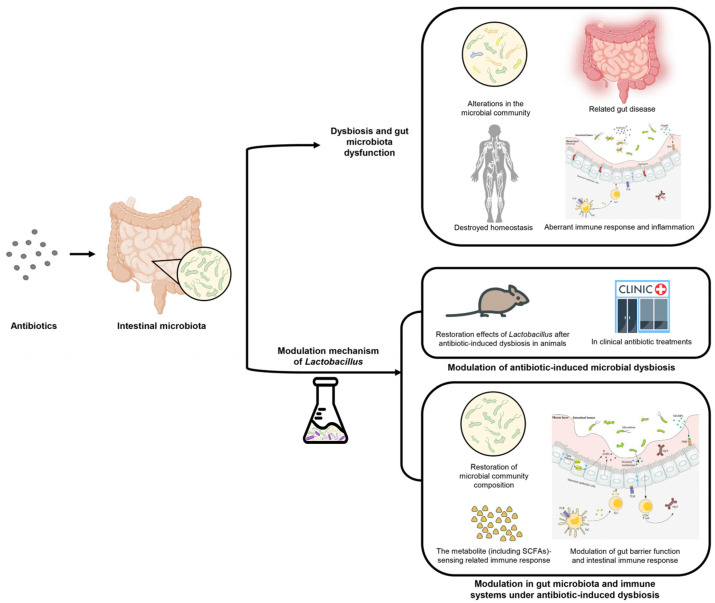
Overview of dysbiosis and dysfunction after antibiotic usage, and modulation mechanisms by diverse *Lactobacillus* strains.

**Figure 2 microorganisms-11-00179-f002:**
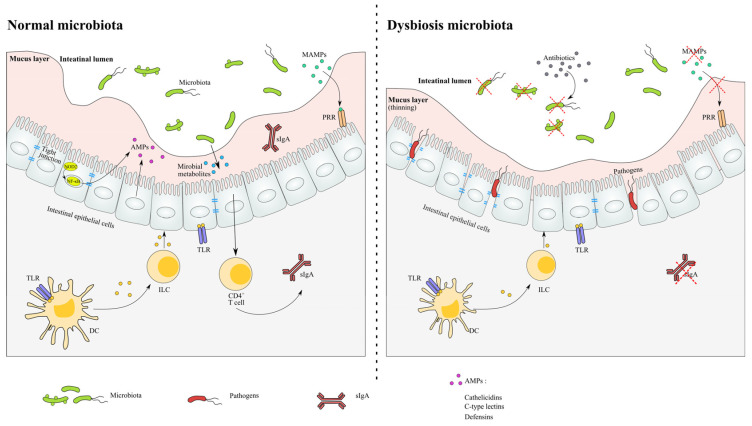
The impact of antibiotics on the microbiota, metabolites and immunity.

**Figure 3 microorganisms-11-00179-f003:**
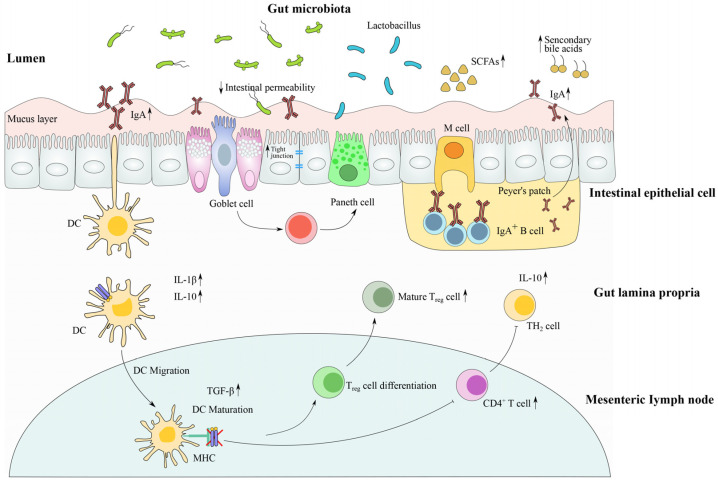
The molecular mechanism of *Lactobacillus* strains in antibiotic-induced dysbiosis.

**Table 1 microorganisms-11-00179-t001:** Associations between β-actin antibiotics and dysbiosis.

Antibiotics	Spectrum	Effects on Microbiota	Risk of Dysbiosis	Outcome of Diseases or Immunity	Ref
Penicillin	Narrow, Gram-positive	Altered the composition, increased *Lachnospiraceae*	Moderate	Increased adiposity and hormone levels in mice	[3]
Penicillin	Narrow, Gram-positive	Altered microbial community composition, reduced *Lactobacillus*, *Candidatus Arthromitus* and *Allobaculum* levels	High	Enhanced the effect of high-fat diet induced obesity and affected ileal genes expression involved in immunity	[28]
Ampicillin	Broad, Gram-positive and some Gram-negative	Decreased microbial community diversity, loss of *Akkermansia*, *Eubacterium*, *Alistipes* and increase of *Staphylococcus*, *Acinetobacter*, *Enterococcus*	Moderate	increased gut permeability, increased the production of inflammatory cytokines including TNF-α, IFN-γ, IL-6, MCP-1 and IL-1β in the ileum	[19]
Ampicillin	Broad, Gram-positive and some Gram-negative	Depleted segmented filamentous bacteria and Gram-positive bacteria	Moderate	Depletion of Th 17 cells	[29]
Amoxycillin	Broad, Gram-positive and some Gram-negative	Changed the microbial composition, Depletion of *Lactobacillus*	Moderate	Decreased expression of MHC molecules and increased mast cell proteases	[30]
Amoxycillin	Broad, Gram-positive and some Gram-negative	Increased *Clostridium clostriforme*, *Eubacterium desmolans*, *Porphyromonas*, *Bacillus mycoides*, *Helicobacter*, *Rumniococcus gnavus* and *R. schinkii*		Not studied	[31]
Amoxycillin	Broad, Gram-positive and some Gram-negative	Decreased richness and Shannon evenness, no significant difference in community structure	Mild	Influenced microbial oxalate-degrading capacity	[2]
Cefixime	Broad, Gram-positive and Gram-negative	Reduction in the diversity of the microbial community and led to decreasing to one preponderant *Firmicutes*	High	Decreased the production of short-chain fatty acids and induced intestinal inflammation	[32]
Cefoperazone	Broad, Gram-positive and Gram-negative	Reduced the total number of bacteria, allowing overgrowth of *Candida albicans*	High	Allergic-airways disease develops after challenge with mould spores	[33]
Cefoperazone	Broad, Gram-positive and Gram-negative	Induced substantial changes in gut microbial community and susceptible to *C. difficile* infection	High	Modified metabolic activity: decreased the levels of glucose, secondary bile acids, free fatty acids and dipeptides	[34]

**Table 2 microorganisms-11-00179-t002:** Effects of *Lactobacillus* strains in antibiotic-induced dysbiosis.

Experimental Design	Strains, Dosage and Duration	Antibiotic	Effects of *Lactobacillus* in the Gut Microbiota and Metabolites	Effects of *Lactobacillus* in the Immune Ecology	Ref
Mice					
C57BL/6 mice aged 6-8 weeks received a chow diet for 7 days with broad-spectrum antibiotics	*L. rhamnosus* GG	Metronidazole, neomycin sulfate andvancomycin	ND	Minimized the decrease expression of butyrate transporter and receptor, and tight junction proteins caused by antibiotic. Decreased GPR109a, SLC5A8, AQP4, and NHE3 transcripts	[64]
Mice receive oral gavage with either cefixime (50 mg/kg) or high-dose cefixime (150 mg/kg)	Cocktails of *L. plantarum*, *L. casei*, *L. rhamnosus*, and *L. helveticus*	Cefixime	Recovered composition of microbiota, enhanced abundance of Firmicutes, decreased Bacteroidetes, Proteobacteria	Decreased serum C-reactive protein, complement 3, and IgG	[32]
Four-week-old male C57BL/6J mice	Cocktails of *L. casei L. plantarum*, *L. rhamnosus*, and *L. helveticus*	Ampicillin	Modulated the microbiota community structure and promoted the abundance of *Akkermansia*	Increased the expression of tight-junction proteins, reducing the production of TNF-α, IL-6, MCP-1, IFN-γ and IL-1β in the ileum and the colon	[19]
Four-week-old male C57BL/6 mice	*L. casei* CGMCC 12,435 (LacC), *L. plantarum* CGMCC 12436 (LacP), or *L. rhamnosus* GG (LacG), respectively	500 mg/kg/day Ampicillin	LacC strain enhanced the alpha diversity and levels of Bacteriodetes and SCFAs	LacC strain enhanced the ileum ZO-1, occluding, down-regulated the expression of NF-κB p65 and modulated the ampicillin-induced inflammatory responses	[65]
Antibiotic-induced microbiota dysbiosis mice with enterococci overgrowth and vancomycin-resistant enterococci persistence	*L. paracasei* CNCM I-3689	1.4 mg/day of clindamycin	Recovery of members of the phylum Bacteroidetes	Increased level of lithocholate and of ileal expression of camp (human LL-37)	[66]
Eleven-week-old male BALB/c mice	3.3 × 10^8^ CFU/mL of either *L. reuteri* 6475 (LR) or *L. rhamnosus* (LGG),	Ampicillin 1.0 g/L and neomycin 0.5 g/L	*L. reuteri* but not *L. rhamnosus* GG reduced the post-antibiotic elevation of the Firmicutes: Bacteroidetes ratio	Increased intestinal permeability, and notably reduced femoral trabecular bone volume (approximately 30%, *p* < 0.01)	[67]
Other animals					
Apis mellifera	Three immunostimulatory *Lactobacillus* strains	Oxytetracycline	Mitigate antibiotic-associated microbiota dysbiosis	Alleviated immune deficits	[68]
Humans					
31 In-patients receiving antibiotics	*L. reuteri*, 2 × 10^8^ CFU, 28 days	Not specific	Decreased AAD among hospitalized adults	ND	[69]
66 subjects screening positive for *H. pylori* infection	*L. rhamnosus*, 1.2 × 10^10^ CFU 7 days	*H. pylori* eradication	Prevent or minimize the gastrointestinal side-effect burden	ND	[70]
Patients treated for infections at an infectious diseases clinic	*L. plantarum* 299v, 1 × 10^10^ CFU, 2 weeks	Not specific	risk of developing loose or watery stools was significantly lower	ND	[71]
Post-antibiotic reconstitution of the gut mucosal host-microbiome niche	Strain cocktails including *Lactobacillus*, *Bifidobacterium*, *Lactococcus* and *Streptococcus*	Ciprofloxacin and metronidazole	Decreased abundance of Clostridiales, recovered microbiome structure	Enhanced expression of ileum REG3G and colon IL1B	[72]
33 participants in patients with an initial mild to moderate *C. difficile* infection	*L. acidophilus* NCFM, *L. paracasei* Lpc-37	Metronidazole; Vancomycinor	Decreased levels of Verrucomicrobiaceae and Bacteroides	ND	[73]
Double-blind, placebo-controlled dose-ranging study, 255 adult inpatients	Mixture of *L. acidophilus* and *L. casei*, 5 × 10^10^ or 1 × 10^11^ CFU, 26 d	One of penicillin, cephalosporin, or clindamycin	Lower antibiotic-associated diarrhea and *Clostridium difficile*-associated diarrhea incidence	ND	[74]
double-blind randomized placebo-controlled trial 85 inpatients	*L. casei*, Shirota, 1.3 × 10^10^ CFU, 12 weeks	Not specific	No improvement	ND	[75]
100 *H. pylori*-positive naive patients	A combination of 2 strains of *L. Reuteri* 2 × 10^8^ CFU, 7 days	*Helicobacter pylori*	*L. reuteri* combination increased eradication rate by 9.1%, and it determines a significant reduction in antibiotic-associated side effects	ND	[76]
Probiotic (*n* = 80) or placebo (*n* = 80) intervention in healthy adults receiving antibiotics	A combination of *L. helveticus* and *L. rhamnosus*, 0·4 × 10^9^ and 7.6 × 10^9^ CFU, 14 days,	Amoxicillin clavulanic acid	Probiotic supplementation reduced the duration of diarrhea-like defecations	ND	[77]
302 hospitalized patients receiving antibiotics	Lactobacillus GG 1 × 10^10^ CFU 14 d	Not specific	*Lactobacillus* GG had no obvious improvement in reducing the rate of occurrence of diarrhea in this sample of 267 adult patients	ND	[78]

## Data Availability

Not applicable.

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
