# Peer review of "Advances in Lactobacillus Restoration for β-Lactam Antibiotic-Induced Dysbiosis: A System Review in Intestinal Microbiota and Immune Homeostasis"

_microorganisms, 2023, doi:10.3390/microorganisms11010179_

Round 1

Reviewer 1 Report

 The manuscript discusses a study on the role of bacteria of the genus Lactobacillus in the treatment of dysbacteriosis of the intestinal microflora caused by β-lactam antibiotics and related diseases.  The relationship between Lactobacillus bacteria and disorders of the host immune system and the interaction between metabolites and immunity when exposed to antibiotics are also being investigated.

 The relevance of this article is justified by the problem of the spread of intestinal dysbacteriosis, due to the increase in various intestinal and chronic respiratory diseases.  The article provides information on the pathogenesis of dysbacteriosis caused by β-lactam antibiotics, homeostasis disorders and changes in the microbial community.

 The article is well structured, and reference used corresponds to the stated topic. The authors presented logical conclusions based on the data obtained during the analysis of studies on this topic.

 However, there are some points that might need clarification to make this review even better:

  -What factors have the greatest impact on the decline in the species diversity of the intestinal microbiota in adults?  Has there been research on this topic?

-  What disturbances of metabolic functions occur in protein metabolism during the development of dysbacteriosis?

- What signaling mechanisms are used by Lactobacillus to suppress other forms of bacteria?

There are also some minor questionable points such as:

Line 19 “relatively imbalanced” -  what do authors mean by “relatively”?

Point 2.1. The antibiotic caused diverse gut bacteria community and destroyed homeostasis – shouldn’t it be formulated in present tense while it is not a narrative story but description of common mechanisms?

2.2.4. Induction of the entire microbial dysbiosis – are you sure about using the term “entire”?

Line 249 – did tiy mean NF-kB? Also, numeration of points here seems excessive (choose 1, 2 or a,b)

Line 308 and mostly studies – in most studies?

Line 453 – present tense will be more suitable

Line 491 the effect exposure – the effect of exposure?

Author Response

Reviewer #1: Comments to the Author

The manuscript discusses a study on the role of bacteria of the genus Lactobacillus in the treatment of dysbacteriosis of the intestinal microflora caused by β-lactam antibiotics and related diseases.  The relationship between Lactobacillus bacteria and disorders of the host immune system and the interaction between metabolites and immunity when exposed to antibiotics are also being investigated.

 The relevance of this article is justified by the problem of the spread of intestinal dysbacteriosis, due to the increase in various intestinal and chronic respiratory diseases.  The article provides information on the pathogenesis of dysbacteriosis caused by β-lactam antibiotics, homeostasis disorders and changes in the microbial community.

 The article is well structured, and reference used corresponds to the stated topic. The authors presented logical conclusions based on the data obtained during the analysis of studies on this topic.

 However, there are some points that might need clarification to make this review even better:

  1. What factors have the greatest impact on the decline in the species diversity of the intestinal microbiota in adults? Has there been research on this topic?

Answer: Thanks for the reviewer’s comments. The topic related to impact on intestinal microbiota by antibiotics in adults, has been investigated widely. Generally, the type of antibiotics (β-lactam, macrolide, quinolones, etc), the dosage and combination, the delivery method (oral, injection or others) are the mainly influence factors on the species diversity of intestinal microbiota. Among them, the category and dosage of antibiotics have the greatest impact on the intestinal microbiota alteration.

  1. What disturbances of metabolic functions occur in protein metabolism during the development of dysbacteriosis?

Answer: According to previous studies, some metabolites, such as amino acids and carbohydrates produced by the colonic microbiota, were found to have decreased in the faecal samples of ampicillin-treated mice. In particular, the SCFA reflected by acetate, propionate and butyrate were significantly decreased or diminished (p < 0.0001) by the antibiotic use (Shi, Y., Kellingray, L., Le Gall, G., Zhao, J., Zhang, H., Narbad, A. et al. 2018. The divergent restoration effects of Lactobacillus strains in antibiotic-induced dysbiosis. Journal of Functional Foods 51: 142-152).

  1. What signaling mechanisms are used by Lactobacillus to suppress other forms of bacteria?

Answer: To our knowledge and related references, Lactobacillus strains may exert SCFAs including acetic acid, propionic acid and butyric acid. SCFAs regulate signalling through GPCRs including GRP43, GPR41 and GPR109A, which are expressed by IECs and immune cells. The GPR109A is a receptor that responds to butyric acid, which prevented colitis by increasing the expression of anti-inflammatory cytokines and inducing differentiation of Treg cells.

There are also some minor questionable points such as:

  1. Line 19 “relatively imbalanced” - what do authors mean by “relatively”?

Answer: It means relative mild imbalanced microbiome. According to the references and previous experiment results, antibiotics could cause imbalanced microbiome in quite different levels. For example, β-lactam antibiotics including ampicillin or amoxicillin could induce mildly alteration of microbiome, meanwhile antibiotic cefixime (β-lactam) or clarithromycin (macrolide) cause disruption of microbiome, and the severity depends on the dose, host and other factors as well. Lactobacillus strains produce positive effect to restore the imbalanced microbiome caused by relative mild antibiotics.

  1. Point 2.1. The antibiotic caused diverse gut bacteria community and destroyed homeostasis – shouldn’t it be formulated in present tense while it is not a narrative story but description of common mechanisms?

Answer: We agree with the reviewer’s comments, and fixed it into “The antibiotics cause diverse gut bacteria community and destroy homeostasis”. Please see line 61 of page 2 in revised manuscript.

  1. 2.2.4. Induction of the entire microbial dysbiosis – are you sure about using the term “entire”?

Answer: As suggested, the term of entire is not suitable to use in this place. We deleted it and changed into “Induction of the intestinal microbial dysbiosis”. Please see line 127 of page 4 in revised manuscript.

  1. Line 249 – did tiy mean NF-kB? Also, numeration of points here seems excessive (choose 1, 2 or a,b)

Answer: Thanks for the reviewer’s comments. It means NF-κB exactly, and the numeration of points has been revised. Please see line 249 of page 2 in revised manuscript.

  1. Line 308 and mostly studies – in most studies?

Answer: The sentence has been revised as suggested into “1010 CFU were applied for administration in most studies”. Please see line 308 of page 8 in revised manuscript.

  1. Line 453 – present tense will be more suitable

Answer: As suggested, the manuscript was revised into present tense as “Lactobacillus strains release metabolic molecules such as short chain fatty acids…”, please see line 453 of page 13 in revised manuscript.

  1. Line 491 the effect exposure – the effect of exposure?

Answer: The sentence has been rephrased as “the effect of antibiotics exposure in disruption of commensal communities…”, and please see line 491 of page 14 in revised manuscript.

Reviewer 2 Report

It's an interesting article.

The descriptions inside the figures should have a higher resolution.

On page 7, in step 1, there appears to be a misprint (NF-?B)

Author Response

Reviewer: 2

Comments to the Author

It's an interesting article.

  1. The descriptions inside the figures should have a higher resolution.

Answer: Thanks for the reviewer’s suggestion. The figures were modified to a higher resolution, and will be submitted in revision.

  1. On page 7, in step 1, there appears to be a misprint (NF-?B)

Answer: As suggested, the misprinted NF-κB was modified, and please see line 297 of page 7 in revised manuscript.